# Intracellular Injection of Brain Extracts from Alzheimer’s Disease Patients Triggers Unregulated Ca^2+^ Release from Intracellular Stores That Hinders Cellular Bioenergetics

**DOI:** 10.3390/cells11223630

**Published:** 2022-11-16

**Authors:** Anna Pensalfini, Abdul Rahim Umar, Charles Glabe, Ian Parker, Ghanim Ullah, Angelo Demuro

**Affiliations:** 1Department of Molecular Biology and Biochemistry, University of California, Irvine, CA 92697, USA; 2Department of Physics, University of South Florida, Tampa, FL 33620, USA; 3Department of Neurobiology and Behavior, University of California, Irvine, CA 92697, USA; 4Department of Physiology and Biophysics, University of California, Irvine, CA 92697, USA

**Keywords:** upregulated calcium signaling, endoplasmic reticulum, amyloid beta (Aβ), Alzheimer’s, inositol 1,4,5-trisphospahte receptors, cellular bioenergetics

## Abstract

Strong evidence indicates that amyloid beta (Aβ) inflicts its toxicity in Alzheimer’s disease (AD) by promoting uncontrolled elevation of cytosolic Ca^2+^ in neurons. We have previously shown that synthetic Aβ42 oligomers stimulate abnormal intracellular Ca^2+^ release from the endoplasmic reticulum stores, suggesting that a similar mechanism of Ca^2+^ toxicity may be common to the endogenous Aβs oligomers. Here, we use human postmortem brain extracts from AD-affected patients and test their ability to trigger Ca^2+^ fluxes when injected intracellularly into Xenopus oocytes. Immunological characterization of the samples revealed the elevated content of soluble Aβ oligomers only in samples from AD patients. Intracellular injection of brain extracts from control patients failed to trigger detectable changes in intracellular Ca^2+^. Conversely, brain extracts from AD patients triggered Ca^2+^ events consisting of local and global Ca^2+^ fluorescent transients. Pre-incubation with either the conformation-specific OC antiserum or caffeine completely suppressed the brain extract’s ability to trigger cytosolic Ca^2+^ events. Computational modeling suggests that these Ca^2+^ fluxes may impair cells bioenergetic by affecting ATP and ROS production. These results support the hypothesis that Aβ oligomers contained in neurons of AD-affected brains may represent the toxic agents responsible for neuronal malfunctioning and death associated with the disruption of Ca^2+^ homeostasis.

## 1. Introduction

Progressive disruption of neuronal Ca^2+^ homeostasis is one of the leading mechanisms of action of the amyloidogenic proteins Aβ as the toxic species in the etiology of Alzheimer’s disease (AD) [1,2,3]. An increase in intracellular content of oligomeric Aβs is believed to play a major role in the early phase of AD as their intracellular rise strongly correlates with AD symptoms, and has been proven to be predictive of cognitive status in AD-affected patients [4,5,6].

Previous results from our lab support the notion that intracellular, as well as extracellular, interactions of Aβ42 oligomers with cell membranes promote cytosolic Ca^2+^ to rise to a toxic level [7,8,9]. Pharmacological and computational studies reveal that applications of synthetic Aβ42 oligomers to different cell types induce a G-protein-mediated stimulation of inositol 1,4,5-trisphosphate (IP**_3_**) production, triggering intracellular Ca^2+^ fluxes leading to a disruption of intracellular Ca^2+^ homeostasis [7,9,10,11]. These findings led us to propose that the abnormal stimulation of IP_3_ overproduction triggered by Aβ42 oligomers may significantly contribute to Ca^2+^ signaling disruptions and neurotoxicity in AD-affected neurons. While there is a widespread agreement on the toxic properties of Aβ42 oligomers, most of this knowledge comes from the use of synthetic A**β** peptides, aggregated in vitro using distinct protocols, generating ambiguity whether endogenous Aβ oligomers may exert similar effects on neurons [2,12,13,14]. The ability to obtain human brain extracts from AD-affected brains, containing high levels of endogenous Aβ aggregates, offer the opportunity to validate these previous findings by investigating their ability to disrupt intracellular Ca^2+^ signaling as observed in neurons of AD-affected brains. Here, we investigate the Ca^2+^-dependent toxicity of selected brain extracts, chosen from postmortem brain samples that had been previously characterized by immunohistochemistry and found to contain high level of OC-positive oligomers. Fluorescence Ca^2+^ imaging experiments were performed using seven different samples of soluble brain extracts: three from normal brain (N1-B11, N2-B11 and N3-B11) and four from AD-affected samples (AD1-B11, AD2-B11 and AD3-TEC and AD3-B11) [4,5]. Experiments were performed by intracellular microinjection of 10 nl of these samples into *Xenopus* oocytes loaded with Ca^2+^ sensitive dye and their ability to trigger cytosolic Ca^2+^ signal was tested. Injections of brain extracts from normal patients evoked little or no detectable fluorescent Ca^2+^ signals. On the contrary, intracellular injection of extracts from AD brains consistently triggered cytosolic Ca^2+^ elevations. These Ca^2+^ signals consisted of both local events and global Ca^2+^ elevations, and showed strong similarity with those evoked after intracellular injection of synthetic Aβ42 oligomers and IP**_3_** [9,15]. Blockage by caffeine (an IP_3_ receptor inhibitor) and OC-antiserum support the ability of endogenous Aβ aggregates contained in the AD brains extracts to trigger cytosolic Ca^2+^ release by uncontrolled activation of IP_3_Rs [16,17,18]. Next, we used computational modeling to quantify the corresponding cytosolic increase in IP**_3_** and Ca^2+^ and the resulting disruption of the normal cell bioenergetics. Overall, our results support the notion that early intracellular rise in Aβ aggregates seen in AD-affected neurons may inflict neuronal cytotoxicity by disrupting intracellular Ca^2+^ homeostasis and cellular bioenergetics.

## 2. Materials and Methods

### 2.1. Oocyte Preparation and Microinjection

We used *Xenopus* oocytes as a model cell preparation, as their large size enables intracellular microinjection of samples [19]. Experiments were performed on stage VI oocytes, injected before imaging with fluo-4-dextran (MW 10,000 D; kD for Ca^2+^ about 3 μM) to a final intracellular concentration of ~40 μM. Oocytes were then placed animal hemisphere down in a chamber whose base was formed by a fresh, ethanol-washed microscope cover glass (Fisherbrand, type-545-M), and bathed in a Ca^2+^-free Ringer’s solution (composition in mM: NaCl, 110; KCl, 2; HEPES, 5; at pH 7.2). A gravity-fed superfusion system allowed exchange of the Ringer’s solution. Oocytes were imaged at room temperature by wide-field fluorescence microscopy using an Olympus inverted microscope (IX 71) (Figure 1A) equipped with a 60X oil-immersion objective, a 488 nm solid state laser for fluorescence excitation and a ccd camera (Cascade 128+: Roper Scientific) for imaging fluorescence emission at frame rates of 10 to 30 s^−1^. Changes in fluorescence intensity were imaged within a 40 × 40 μm region in the animal hemisphere of the oocyte and measurements are expressed as a ratio (ΔF/Fo) of the change in fluorescence at a given region of interest (ΔF) relative to the resting fluorescence at that region before stimulation (Fo). Mean values of Fo were obtained by averaging over several frames before stimulation using MetaMorph (Molecular Devices). Fluorescent traces were exported to Microcal Origin version 6.0 (OriginLab, Northamptom, MA, USA) for analysis and graphing. Microinjection of 10 nl of brain extract samples into oocytes was performed using a Drummond nanoinjector mounted on a hydraulic micromanipulator. A glass pipette (tip diameter of 8–10 μm) was filled with samples and inserted vertically down through the entire oocyte to a pre-established position with the tip positioned 6 to 8 μm inward from the plasma membrane and centered within the image field.

### 2.2. Patient Brain Tissue Selection

Frozen brain tissue was obtained from the UCI Alzheimer Disease Research Center (ADRC). The cognitive status of subjects enrolled in the ADRC was assessed with mini mental status exam (MMSE). As a standard protocol for ADRC autopsy cases, Braak & Braak neurofibrillary tangle and plaque staging was evaluated [20]. Our objective was to test brains samples with high content of OC positivity to antiserum. We selected tissues from the frontal cortex (Brodmann’s Area11; B11) from two different AD-affected individuals, and from a third individual from which both B11 and trans-entorhinal cortex (TEC) samples where available (Table 1). These subjects were widely characterized in our previous studies for their immunoreactivity with various conformation-dependent antibodies, such as OC and mOC78, which correlated with cognitive decline, tangle stage and plaque pathology, as well as with early intracellular/intranuclear aggregates build up at intermediate stages of plaque pathology (plaque stage A–B), respectively [4,5]. Table 1 lists the clinical and pathological details of the cases used in this study. The stock solution for all the brain extract was estimated to be 1 μg/ml for Aβ contents in PBS. For comparison with our previous investigations using synthetic Aβ42 oligomers most of the experiment reported here were performed injecting 1 μg/ml, except for the experiments where OC antiserum was used to inhibit endogenous Aβs activity where they were used with OC of 0.5 μg/ml.

### 2.3. Human Brain Sample Preparation and Dot Blot Analysis

Frozen tissues from the B11 area or trans-entorhinal cortex (TEC) were processed as previously described [5]. Briefly, the brains were weighed, diced, and homogenized in ice-cold PBS (4:1 PBS volume/brain wet weight), 0.02% NaN_3_, pH 7.4, supplemented with protease inhibitor cocktail (Roche). The samples were ultracentrifuged at 100,000× *g* for 1 h at 4 °C. The PBS soluble fraction was collected, aliquoted and stored at −80 °C for future testing.

For dot blot analysis, 2 μg of samples was spotted in duplicate onto nitrocellulose membrane and allowed to air dry. After blocking the non-reactive sites with 10% nonfat dry milk in low-Tween TBS-T, (20 mM Tris, 137 mM NaCl, 0.01% Tween 20 pH 7.6), the blots were incubated overnight at 4 °C in primary antibody at the following dilutions: OC 1:10,000, A11 1:2000, 6E10 (Covance, Princeton, NJ, USA) 1:2000. The blots were then incubated with goat-α-rabbit (for OC and A11) and goat-α-mouse HRP conjugated secondary antibody (Jackson Immune Research 1:12,000), washed three times for 5 min, followed by ECL detection. Stock solution for OC was 0.4 μg/ml in PBS solution.

### 2.4. Computational Methods

Equations modeling cytosolic Ca^2+^ dynamics and mitochondrial function, and numerical methods are described in detail in Appendix A.

## 3. Results

### 3.1. Brain Extracts from AD Patients Display High Content of OC-Positive Aβs

Of relevant interest in understanding the role of soluble Aβ (Aβs) oligomers in the etiology of AD is to uncover the prevailing type of aggregates that more closely associate with the development of the symptoms and the associated molecular mechanisms [21]. Previous work from various labs, including ours, has demonstrated that the time-dependent accumulation of intracellular Aβ oligomers is strictly correlated with the progression of the symptoms over time. In addition, intracellular Ca^2+^ rise observed in neurons during aging has been linked to the action of specific oligomeric forms of Aβs.

In this work we selected samples of postmortem brain extracts from three normal and three AD-affected patients which were screened for their affinity to selected antibodies for different types of Aβ aggregates [4,5]. Figure 1A shows dot blot analysis of human soluble PBS fractions probed with OC, A11, 6E10 and 4G8 antibodies. OC positive oligomers, but not A11, 6E10 nor 4G8 immunoreactivity, was increased in AD brain samples compare to normal samples. In Figure 1B, a comprehensive plot shows the calculated fold changes in immunoreactivity to each antibody. The plot shows that very low content of OC positive Aβs oligomers was found in samples N1-B11, N2-B11 and N3-B11 from normal brains, whereas high content of OC positive oligomers was detected in all the samples from AD affected brains AD1-B11, AD2-B11 with particular high immunoreactivity in AD3-B11/TEC samples.

### 3.2. Intracellular Injection of Brain Extracts Induces Local and Global Cytosolic Ca^2+^ Fluxes

To examine the ability of human brain extracts to trigger Ca^2+^ mobilization from intracellular stores, we used fluorescent Ca^2+^ imaging in *Xenopus* oocyte as a model system. The experiments described here followed similar approaches we previously applied to investigate the Ca^2+^ toxicity of synthetic Aβ42 oligomers. The stock solution of brain extracts used in this work were estimated to contain up to 1 μg/ml of Aβ in PBS solution. Intracellular injection of 10 nl samples at this concentration triggered fast and robust fluorescent responses. Further, we observed that injection of 10 nl of lower concentrations, down to 0.3 μg/ml dilutions, retained the ability to consistently trigger Ca^2+^ signaling (data not shown).

Intracellular microinjection of 10 nl samples from AD brain extracts evoked potent cytosolic Ca^2+^ mobilization of different spatiotemporal patterns, ranging from repetitive local brief events such as blips and puffs to global repetitive Ca^2+^ waves covering the entire image field. Conversely, injection of 10 nl of brain extracts from normal individuals failed to promote either local or global Ca^2+^ fluxes. In Figure 2, we show results from selected series of single experiments depicting the spatiotemporal evolution of the Ca^2+^-dependent fluorescent signals evoked by injection of different brain extracts.

In Figure 2A, the top trace depicts the Ca^2+^-fluorescent signal evoked in response to intracellular injection of 10 nl of AD3-TEC sample at the time indicated by the arrow in the line-scan. Below is the corresponding line-scan (kymograph) image, showing temporal evolution of the fluorescent signal originated by measuring changes in fluorescence intensity along a single line scan (25 μm long and 5μm wide) properly positioned above the centroid of the fluorescent signal. The panel displays the signal evolution over time as a pseudo-colored representation where warmer colors indicate increase in fluorescence ratio intensities. Fluorescent transients with different amplitudes and duration are evident along the line-scan, reminiscent of the hierarchical organization of IP3-mediated Ca^2+^ signaling. Figure 2B,C illustrate the corresponding fluorescent response to injection of AD3-B11 and AD1-B11 samples showing large variability in time and amplitudes of the fluorescent responses. This variability was shared among all the tested AD-affected samples, independently from the donor or brain region (B11 or TEC). Figure 2D shows a typical response obtained after intracellular injection of a 10 nl sample from normal brain, representative of 12 trials which consistently failed to give detectable fluorescent responses.

In Figure 3A a selected number of fluorescent traces (black) from 23 different injections including a proportional number of traces from each sample of AD-affected brain extracts are shown. Traces are aligned to the first initial response for each injection and are not indicative of time delay of the fluorescence response after injection. Control experiments using samples from normal individuals, consistently failed to evoke fluorescent response. Selected fluorescent traces obtained from experiment using different samples from control brains are superimposed and plotted together in Figure 3B. The overall results obtained in this series of experiments in outlined in Figure 3D, where the number of responding cells from all the experiments are reported as percent of the total cells tested. In all the experiments performed, samples from AD patients consistently triggered fluorescent responses independent of the gender of the donor or the specific area of the brain from which the sample originated, whereas comparable samples from normal donor of equivalent age consistently failed to evoke significant fluorescent responses. Only a single response consisting of a single localized brief fluorescent transient (puff) was detected throughout the imaged field that we attributed to endogenous spontaneous Ca^2+^ signaling. We applied similar approach for screening all the trials performed with active responses classified as fluorescent responses consisting of at two separated sites with repetitive transients at each site.

### 3.3. Conformation-Dependent OC-Antiserum Strongly Inhibits Ca^2+^ Mobilization by AD-Extracts

To confirm that the soluble Aβ oligomers contained in the AD-affected samples were responsible for the Ca^2+^ dependent fluorescent transients observed after injection, we used the conformation-specific OC antiserum to neutralize the activity of endogenous Aβs oligomers. To this aim, we performed a series of experiments where samples from AD brains were pre-incubated for 24 h with OC-antiserum prior to injection. In parallel, equivalent aliquots of corresponding samples were incubated without addition of antibody. In Figure 4A top-left, the trace depicts the temporal evolution of the fluorescent signal in response to intracellular injection of 10 nl AD3-TEC sample at the time indicated by the arrow. In this specific experiment, within 10 s after injection a robust, fast rising fluorescent response was observed which covered a large portion of the imaged field (40 × 40 μm). The trace was obtained by measuring the average intensity within a region of interest covering the footprint of the fluorescent signal. The time delay in the fluorescent response was very common in all the experiments and we never observed fluorescent response with a delay longer than 30–40 s.

We then performed parallel experiment using the corresponding sample which had been previously incubated with OC antiserum. Top-right trace in Figure 4A depicts the averaged intensity of the fluorescent signal recorded after injection of 10nl of AD3-TEC+OC. The sample was injected at the time indicated by the arrow and with ~30 s delay a pulse of UV was applied as to confirm functional capability of the IP3 signaling system. Similar experiments for samples AD3-B11 and AD2-B11 are reported in the middle two traces and bottom two traces of Figure 4A, respectively. The cumulative results of this set of experiments are reported in Figure 4B. Each column in the plot represents the number of responding cells as percentage of the total cell tested as indicated on top of each column.

### 3.4. Ca^2+^ Fluxes Triggered by Endogenous Aβs Are Initiated by IP_3_-Mediated Intracellular Liberation

The close resemblance between the temporal evolution of the fluorescence responses triggered by intracellular injection of the AD-samples used in this investigation and IP_3_-mediated Ca^2+^ signaling suggests that intracellular AD brain extracts promote the liberation of Ca^2+^ from the ER stores via opening of IP_3_Rs. Moreover, following a similar approach, we have previously reported that intracellular injection of synthetic Aβ42 oligomers into Xenopus oocytes triggers cytosolic Ca^2+^ release through a mechanism involving IP_3_ overproduction. In Figure 5, we show a comparative analysis of localized Ca^2+^ events triggered by AD brain extracts (Figure 5A), synthetic Aβ42 oligomers (Figure 5B), and by injection of 10 nM IP_3_ (Figure 5C). In these analyses only selected experiments where local Ca^2+^ events were triggered are included.

In Figure 5A, the top image is a kymograph illustration of puff-like events occurring at the same location, promoted by intracellular injection (10 nl) of samples from AD-affected brains. Image is generated from the measurement of fluorescence along a line (*y*-axis; 5 μm long and 2 μm wide) positioned above the puff`s footprint in the video record, with time running left to right along the *x*-axis. Increasing fluorescent signals are represented by warmer colors as indicated by the color bar and by increasing height of each pixel. Below, the left plot is generated by the overlapping of 10 randomly selected traces aligned to the initial rise of the fluorescent trace. Inset is generated by averaging all the puffs traces. Center and right plots in Figure 5A, respectively show distributions of puff durations (measured at half-maximal amplitude) and maximum amplitudes obtained from measurement of 287 puffs from 6 oocytes from experiments carried out with different brain extracts.

Figure 5B shows analysis of fluorescent events evoked by intracellular injection of 10 nl of synthetic Aβ42 oligomers (1 μg/ml). Top image illustrates repetitive puffs occurring at the same location in the image field. Below, the left plot is generated by the overlapping traces from 10 selected puffs. The trace in the inset is generated averaging the 10 traces in main plot. Center and right plots are the distribution of the measured puff durations and amplitudes. Figure 5C illustrates comparative analyses of the fluorescent transients generated directly by intracellular injection of 10 nl of IP_3_ (10 nM).

The top image displays three fluorescent transients occurring at the same location, closely resembling the local transients observed when synthetic and endogenous Aβ oligomers are injected. Overlapping of the puff-like profiles is shown in left plot and inset, shows the resulting averaged intensity profile. Corresponding distribution of the average events duration and amplitudes are shown in the center and right plots, respectively.

Overall, this comparative investigation supports our hypothesis that the elementary Ca^2+^ events triggered by brain extract from AD patient are similar to those generated by the synthetic Aβ42 oligomers and by IP_3,_ and involve the opening of IP_3_Rs [22,23].

To further strength our hypothesis, we examined the effect of caffeine, known to act as a reversible antagonist of IP_3_Rs on the Ca^2+^ fluxes triggered by endogenous Aβ oligomers [24]. Moreover, the ability of caffeine to freely permeate the oocyte’s membrane and the absence of ryanodine receptors, make caffeine an ideal agent to inhibit IP_3_Rs in Xenopus oocytes [16,24].

In these experiments the oocytes were peeled (the vitelline membrane was removed) to increase the adhesion of oocytes to the coverslip, ensuring recordings at the same exact location during application and after the washing of caffeine from the bathing solution. In Figure 6A a set of three traces are shown, reporting fluorescent intensity recorded at the same location after injections of 10 nl of AD3-TEC sample at the time indicated by the arrow. In control conditions (left panel) an initial spike after ~5 s was followed by a slower change in fluorescent signal. Perfusion of caffeine in the bathing solution was performed by gravity driven fashion with a flux of about 0.4 ml/s, able to fully exchange the volume of the chamber with 10 s. To ensure full concentration of caffeine in the recording chamber, perfusion was left effective for about 2 to 3 min and then stopped. After 5 min incubation with caffeine, with the injecting needle left unmoved, a second injection of 10 nl of AD3-TEC sample was delivered and the corresponding recording of the fluorescent signal is reported in the central pane of Figure 6A, showing the ability of caffeine to completely abolish Ca^2+^ fluorescent transient triggered when AD3-TEC was injected alone. Confirming the reversibility of caffeine’s effect on the IP_3_Rs, the Ca^2+^ response after injection of 10 nl of AD3-TEc was completely restored after extensive washing-of caffeine from the bathing solution (right panel, Figure 6A). Similar results were obtained from experiments injecting 10 nl AD3-B11 into oocytes with and without caffeine (Figure 6B). In the experiment shown in Figure 6C, scattered local transient puff were visualized throughout the membrane patch. The application of caffeine completely abolished the activity at each puff site. Puffs activity was restored after removal of caffeine as shown in Figure 6C right panel, where 10 nl of AD1–B11 sample was injected at the same location.

### 3.5. Computational Quantification of the Ca^2+^ Released and the Corresponding Concentration of IP_3_ Overproduction by Cytosolic Injection of Brain Extracts

To quantify the concentration of IP_3_ generated and the resulting concentration of Ca^2+^ released from intracellular stores in response to intracellular injection of brain extracts from normal and AD-affected brains, we fitted the whole-cell model discussed in the Section 2 to the observed fluorescence traces. Sample fits to the fluorescence traces are shown in Figure 7A–D, where red lines represent the fluorescence changes estimated by the model.

Observed fluorescence changes (black) are shown for comparison. The blue curves represent the estimated change in cytosolic Ca^2+^ concentration in response to brain extracts injection. The time-traces for Ca^2+^ and IP_3_ concentrations are integrated to estimate the total Ca^2+^ and IP_3_ concentrations generated by brain extracts from control brain and three different AD brains, and are shown in Figure 7E,F, respectively. Plots in Figure 7E,F clearly show a significantly larger IP_3_ generated and consequently higher Ca^2+^ released from intracellular stores due to extracts from AD patients as compared to extracts from control brains.

### 3.6. Ca^2+^ Dependent Disruption of Bioenergetics

Next, we used our model to estimate the bioenergetic cost of the aberrant Ca^2+^ rises due to brain extracts. Specifically, we feed the Ca^2+^ traces estimated above to a model for mitochondrial bioenergetics to estimate the changes in mitochondrial Ca^2+^ concentration, cell’s ATP, and ROS levels as detailed in the Section 2. Sample traces for these three variables in cells injected with control and AD2-B11 extracts are shown in Figure 8A,C,E, respectively.

A summary of changes in mitochondrial Ca^2+^ concentration, ATP and ROS obtained from fits to three experiments are shown Figure 8B,D,F, respectively. Both the traces and bar plots reveal a significant drop in ATP and rise in ROS levels in cells injected with extracts from AD-affected brains.

## 4. Discussion

A great deal of progress has been made in recent years to better understand the involvement of Aβ in the progression of Alzheimer’s disease [25,26,27]. A widely accepted hypothesis implicates the oligomeric forms of Aβ as the main toxic agent triggering an uncontrolled elevation of cytosolic Ca^2+^ to a poisonous level, with consequent impairment of normal cellular functioning and final death [1,2,7,14,28]. Using Ca^2+^-dependent fluorescent imaging we had previously proposed two distinct mechanisms by which Aβ42 affects intracellular Ca^2+^ homeostasis. One, by forming self-gating Ca^2+^-permeable pores allowing uncontrolled influx of extracellular Ca^2+^ into the cytosol; and a second mechanism involving the release of Ca^2+^ from intracellular stores through the opening of IP_3_R channels [7,8]. These two mechanisms have been widely investigated by various groups using many different approaches [29,30,31,32,33]. Nevertheless, most of these investigations have been carried out using synthetic Aβ peptides, leading to ambiguity as to whether the proposed mechanisms of action are also applicable to endogenous Aβ oligomers found in neurons of AD-affected brains [34]. Here, we report the ability of brain extract samples displaying a high content of Aβ oligomers to evoke cytosolic Ca^2+^ release upon intracellular injection, closely mirroring the effect of synthetic Aβ42 oligomers in analogous experiments. The samples consist of homogenate extracts from postmortem brain tissues from frontal cortex of donors with clinical and pathological details shown in Table 1. Three samples from the frontal cortex (B11) of normal individual were used as control and compared with four samples (three B11 and one TEC) from three individuals exhibiting the typical hallmarks of AD-affected brains (Table 1). The immunological characterization shown in Figure 1A reports dot blot analysis of samples probed with conformation-specific OC and A11 antibodies and sequence-specific 6E10 and 4G8 antibodies [35,36,37,38,39]. All the samples we tested displayed low immunoreactivity to A11, 6E10, and 4G8, whereas high immunoreactivity was detected for OC antiserum (Figure 1B). Quantitative analysis of the dot blot data show significant levels of OC immunoreactivity in the AD samples compared to normal subjects. Particularly, AD3 samples for both B11 and TEC regions displayed a higher content of OC-positive Aβ oligomers. Moreover, AD3 brains showed increased plaque stage as compared to AD1 and AD2 (Table 1). Intriguingly, all the AD samples displayed low immunoreactivity to A11, indicating that the endogenous Aβ oligomers contained in the brain samples had a specific fibrillar oligomer aggregation state, equivalent to that acquired by the synthetic Aβ42 oligomers in our previous preparations [8].

As shown in Figure 2, intracellular injections of 10 nl samples from AD3-TEC, AD3-B11 and AD1-B11 donors, all triggered cytosolic Ca^2+^ fluxes ranging from local events to global events involving large areas in the image field. Conversely, 10 nl N2-B11 sample from control brain, failed to trigger any significant response similarly to the effect of 10 nl injection of PBS (Figure 3D). The absence of Ca^2+^ in the bathing solution points to a release of Ca^2+^ from intracellular stores. High variability in amplitudes and temporal evolution of the fluorescent signals, including local and global events evoked by all the AD-affected samples (Figure 3A) closely resembles the fluorescent responses obtained after intracellular injection of synthetic Aβ42 oligomers and agonist IP_3_ [9,15].

Intracellular elementary events in the IP_3_ pathway signaling are the building blocks by which local and global Ca^2+^ signaling are coordinated [40]. The strong similarity of the elementary events evoked by AD brain extracts with those evoked by the direct injection of the agonist IP_3_ supports the hypothesis of Ca^2+^ release from intracellular stores through activation of IP_3_Rs. This proposition is reinforced by experiments where we challenged the ability of brain extracts to trigger Ca^2+^ events by pre-incubating oocytes with 10 mM caffeine (Figure 5). Caffeine acts as a reversible, membrane-permeant competitive antagonist of the IP_3_R [16,24].

In comparable conditions, i.e., pre-treatment with caffeine and intracellular delivery of Aβ42 oligomers, similar result was reported using SH-SY5Y cells a model system [11]. Furthermore, the authors observed an additional IP_3_-indipendent Ca^2+^ release from the ER, suggesting that Aβ42 oligomers may trigger a Ca^2+^ leak from the ER which does not depend upon a direct interaction with IP_3_Rs; a behavior that we did not observe in the time range of our experiments. However, in similar experimental conditions, including zero extracellular Ca^2+^ and caffeine pre-treatment, we did observe cytosolic fluorescent Ca^2+^ transients after longer time following intracellular injection of Aβ42 oligomers (unpublished data; Demuro and Parker). Considering the amplitudes and temporal evolution of these Ca^2+^ transient we speculate that over time, interaction of Aβ42 oligomers with the ER membrane may trigger formation of Aβ42 pores in the ER membrane causing the Ca^2+^ leak described by those authors [11].

The specific involvement of the endogenous Aβ oligomers as the agent responsible for the Ca^2+^ mobilization by the brain extract is further strengthened by the ability of the conformation-specific antiserum OC to specifically decrease or completely neutralized the ability of the brain extracts to trigger Ca^2+^ fluxes (Figure 4). The fact that neither the sequence-specific 6E10 nor 4G8 antibodies displayed any detectable immunoreactivity to any of the AD brain extracts suggests that most of the Aβs present in the samples were in a specific aggregated form, and not as random coil monomers [17]. Moreover, the finding that among the two non-sequence specific/conformation specific antibody, only OC but not A11 showed high immunoreactivity in all the samples suggests that OC-positive Aβ fibrillar oligomers are present in substantial amount in the cortex of AD affected brains [4,5,17]. Using pharmacological and computational investigation, we previously proposed that synthetic Aβ42 oligomers affect cell functioning through the overproduction of IP_3_ by stimulation of the G-protein complex to trigger uncontrolled release of Ca^2+^ through IP_3_Rs [10,41]. Intriguingly, those experiments were carried out using synthetic Aβ42 oligomers with high immunoreactivity to OC antiserum, implying that similar toxicity may be mediated by endogenous intracellular Aβ oligomers in neurons of AD affected brains. Interesting, the ability of OC to react with Aβ aggregates that range from a few aggregated peptides to up to 40–50 peptides suggests that more than the size of the aggregate the key aspect is their specific aggregation conformation. These results support the hypothesis that the intracellular rise of endogenous amyloid oligomers observed in neurons of AD-affected brains represent the toxic agents responsible for neurons malfunctioning and death, associated with the disruption of neuronal Ca^2+^ homeostasis.

Analogous conclusions were recently made from experiments using site-specific overexpression of Aβ42 using adeno-associated viral (AAV) constructs to promote either intra or extracellularly expression of oligomeric Aβ42 [42]. From this study the authors concluded that overexpression of Aβ42 oligomer in both compartments can compromise normal synaptic functions and point out that early intracellular accumulation of Aβs may represent the initial trigger of neuronal malfunctioning which can be exacerbated by the extracellular rise in Aβs oligomers [42]. In line with our experimental results, a mathematical model investigating the initial trigger causing Aβ42 toxicity supports the notion that a G-protein coupled stimulation of the PLC pathway by Aβ42 oligomers is responsible for the triggered overexpression of IP_3_ and the observed consequent cytosolic Ca^2+^ rise [43]. Interestingly, a further dose-dependent computational investigation supports a mixed mechanism in which the presence of higher Aβ42 concentrations may also alter the sensitivity of IP_3_Rs to low- or sub-threshold IP_3_ levels and in turn trigger local and global Ca^2+^ signaling events [10].

We have previously proposed that the downstream effect of cytosolic Ca^2+^ rise triggered by synthetic Aβ42 oligomers may involve disruption of normal mitochondrial functioning [41]. Here, after estimating the concentration of IP_3_ generated and the corresponding concentration of Ca^2+^ released through IP_3_Rs caused by AD-affected brain extracts, we fed the resulting time-traces of Ca^2+^ concentration to our computational model for cell bioenergetics to assess the ATP and ROS produced. Our simulations show a significant drop in ATP and rise in ROS due to extracts from AD-affected brains, mostly of short duration. However, we believe that over extended periods, the pathological Ca^2+^ signals and consequently the low ATP and high ROS levels would make the cell progressively vulnerable, eventually leading to the observed synaptic dysfunction and cell’s demise.

In line with this interpretation are the results from our recent study using a biophysical model of hippocampal synapses to investigate the effect of aberrant Ca^2+^ signaling on neurotransmitter release [44]. The model predicts that enhanced Ca^2+^ release from the ER increases the probability of neurotransmitter release in AD-affected neurons. Moreover, over very short timescales (30–60 ms), the model exhibits activity-dependent and enhanced short-term plasticity in AD, indicating neuronal hyperactivity. Similar to previous observations in AD animal models, pathological Ca^2+^ signaling increases depression and stimulus desynchronization causing affected synapses to operate unreliably [44].

We remark that while our bioenergetics model has been developed over many years, is based on extensive data, and has been successfully applied to different cell types including oocytes [41] and mammalian cells [45], it has some limitations. For example, the model assumes that endogenous Aβ oligomers only affect mitochondrial function through upregulated cytosolic Ca^2+^ without directly interacting with the mitochondrial membrane. Furthermore, the model ignores the contribution of ryanodine receptors to cytosolic Ca^2+^ concentration and assumes that Aβ oligomers only cause the release of Ca^2+^ from the ER through IP_3_Rs. While this assumption is valid for Xenopus oocyte, ryanodine receptors are shown to play a key role in Ca^2+^ signaling disruptions in AD-affected neurons [46,47]. Thus, the cell-specific differences in Ca^2+^ signaling must be taken into consideration while interpreting the results from our model.

Nevertheless, we believe that our experimental observations and model simulations make a strong case for disruptions in intracellular Ca^2+^ homeostasis and consequently in cell bioenergetics as plausible mechanisms for the toxicity of endogenous Aβ oligomers in AD affected brains.

## Figures and Tables

**Figure 1 cells-11-03630-f001:**
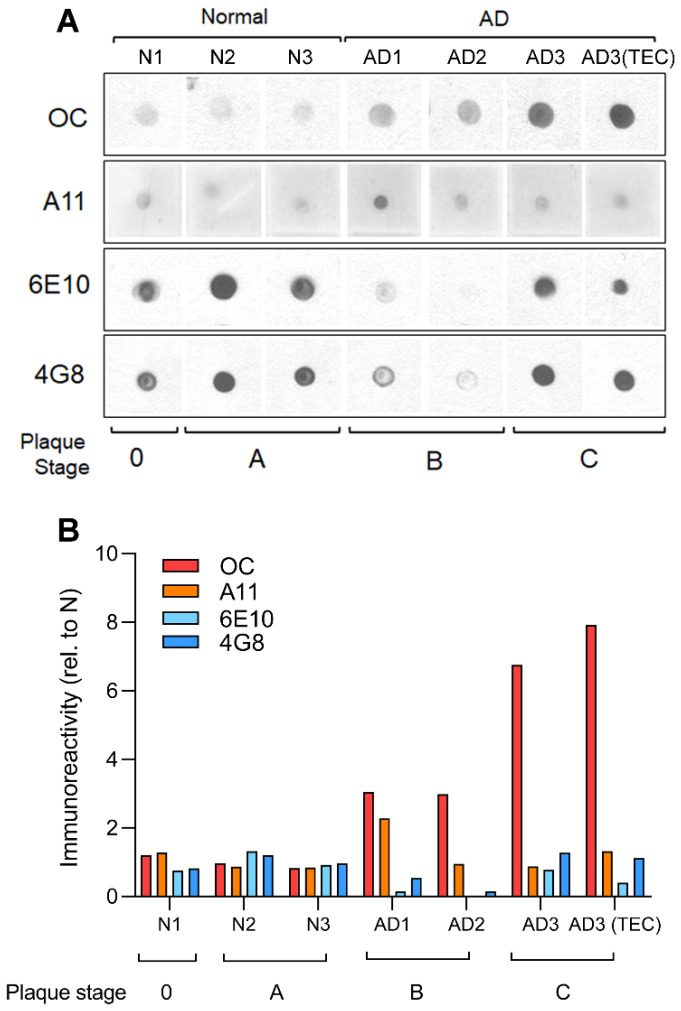
Immunological characterization of human AD brain extracts reveals high content of OC positive fibrillar oligomers. Dot blot analysis (**A**), and respective quantification (**B**), of human soluble PBS fractions from the B11 and TEC regions of normal and AD patients probed with OC, A11, 6E10 and 4G8 antibodies. OC-positive fibrillar oligomers, and not A11-positive prefibrillar oligomers, were increased in AD brain sample with increasing plaque stage pathology. The apparent decrease in 6E10 and 4G8 immunoreactivities in the AD patients with intermediate plaque pathology (stage B) suggests that at this stage Aβ undergoes a conformational change that is not recognized by 6E10 or 4G8, but that can be readily detected by OC.

**Figure 2 cells-11-03630-f002:**
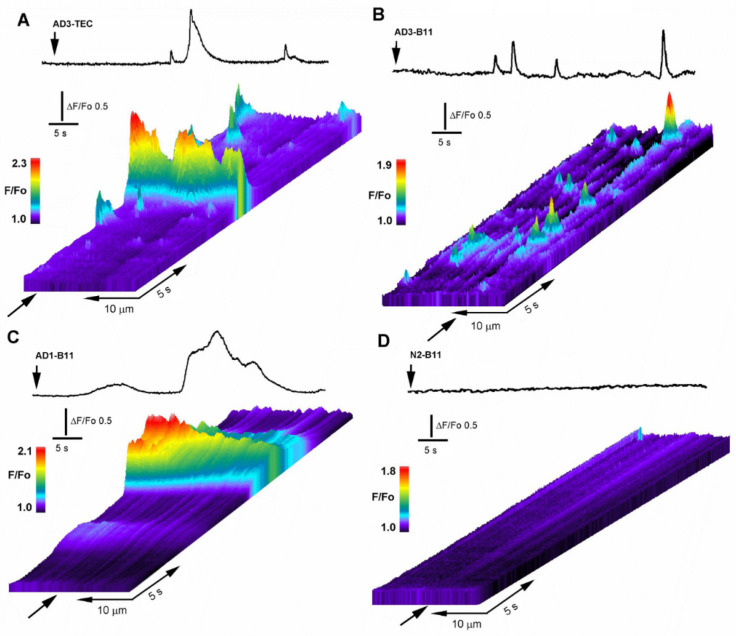
Intracellular injection of OC-positive brain extracts trigger local and global intracellular Ca^2+^ transients. (**A**) Line-scan (kymograph) images illustrating spatiotemporal pattern of fluorescence Ca^2+^ signals recorded from an oocyte injected with 10 nl of AD3-TEC. Top trace shows fluorescence signal monitored from a small region along the line-scan positioned as marked by the arrow in the line-scan image. Brain extract was injected at the time indicated by the vertical arrow. Increasing fluo-4 pseudo-ratio signals are represented by warmer colors as depicted by the color bar and by increasing height at each pixel. (**B**,**C**) corresponding fluorescence traces and kymograph images obtained in oocytes injected with AD3-B11 and AD2-B11, respectively. (**D**) fluorescent signal measured across the image field and kymograph image of the fluorescent background in an oocyte injected with brain extract from a normal subject (N1-B11).

**Figure 3 cells-11-03630-f003:**
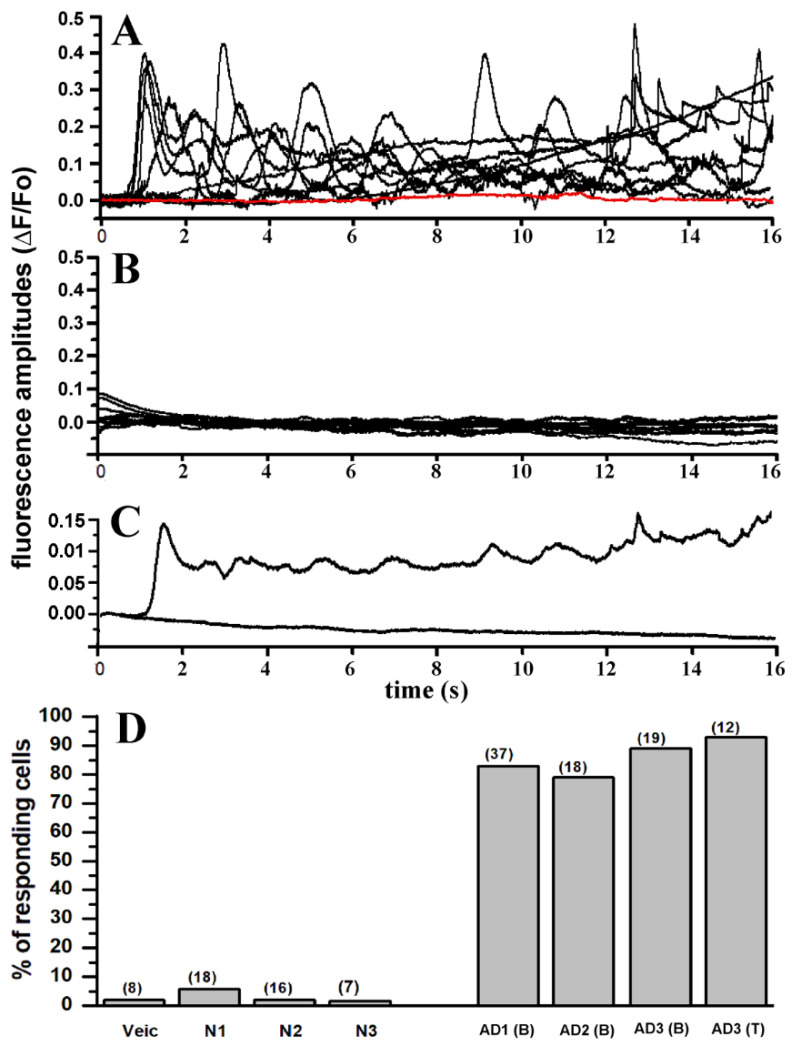
Intracellular injection of brain extracts from AD patients evoked large and consistent intracellular Ca^2+^ signals which are absent upon injection of brain extracts from non-AD patients. (**A**) Superimposed traces represent fluorescent signals recorded from 10 different oocytes in response to intracellular injection with (10 nl) of OC-positive brain extract from patient AD1 (**B**) (see Table 1 for description). Red trace indicates fluorescent background in a non-injected oocyte. (**B**) Superimposed fluorescent traces from 10 different oocytes recorded after injection with brain extracts from non-AD patient. (**C**) Plot shows averaged fluorescent traces from (**A**,**B**). (**D**) Bar graph reporting the number of oocytes showing activation of fluorescence Ca^2+^ signals after intracellular injection of 10 nl of: PBS (vehicle), N1, N2, N3 brain extract from normal patients (see Table 1), and brain extract from AD affected patients–AD1-B11, AD2-B11, AD3-B11 and AD3-TEC. Each column represents the number of responding oocytes expressed as percentage of the total number of oocytes tested for each group. Number of oocytes for each group in indicated by the number on top of each column.

**Figure 4 cells-11-03630-f004:**
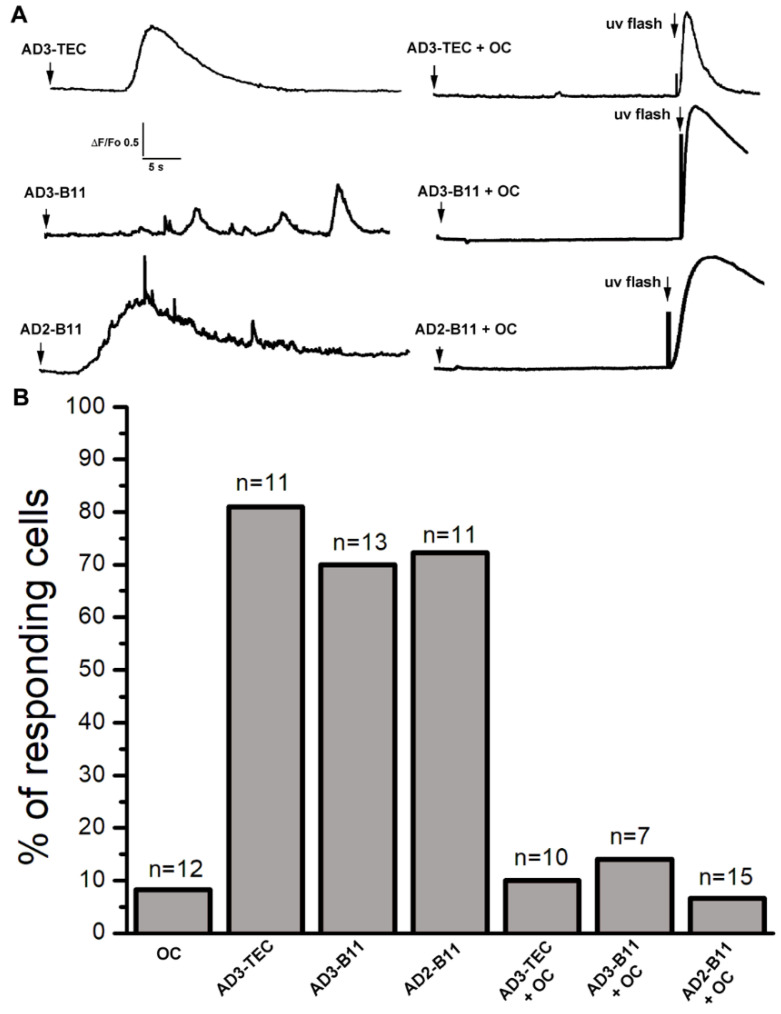
Incubation of AD brain extracts with OC antiserum strongly inhibits their ability to activate Ca^2+^ signals. (**A**) Top-left trace shows a Ca^2+^ fluorescent transient triggered by injection of 10 nl of AD3-TEC (Aβ 0.5 μg/ml) at the time indicated by the arrow. Ca^2+^ response was abolished by preincubation of AD3-TEC sample with OC antiserum (OC 0.2 μg/ml; Aβ 0.5 μg/ml). At the end of the recording a UV flash was given to trigger photorelease of IP_3_ as a control. The middle traces show fluorescence Ca^2+^ signals recorded from oocytes injected with AD3-B11 and corresponding fluorescence record obtained after injection of pre-incubated AD3-B11 with OC antiserum. Bottom traces are obtained from oocyte injected with AD2-B11 (**left**), and after incubation with OC antiserum (**right**). (**B**) Cumulative results showing percent of responding cells for each sample. Columns, represent numbers of oocytes responding to intracellular injection from samples indicated at the bottom of the graph. The data are expressed as percentage of total number of oocytes tested for each group, and n at the top of each column represent the total number of oocytes tested. Oocytes were injected with OC antiserum alone as control.

**Figure 5 cells-11-03630-f005:**
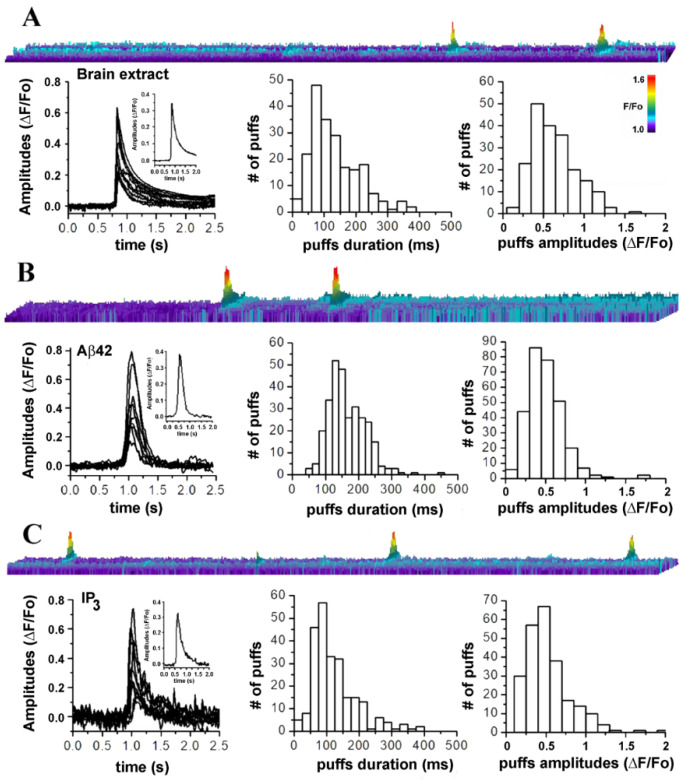
Amplitude and temporal evolution of local Ca^2+^ signals triggered by intracellular injection of brain extracts from AD patients closely resemble those triggered by synthetic Aβ42 oligomers and by IP_3_. (**A**) Top image is a line-scan (kymograph) representation of local intracellular Ca^2+^ events, evoked by intracellular injection (10 nl) of AD3-TEC. Below, left plot shows superimposed traces from 10 different events (puffs). Inset is the resulting averaged profile. Center and right plots show distributions of events duration (mean duration 132.30+/−0.1 ms) and amplitudes (mean amplitude ΔF/Fo 0.378+/−0.058) measured from 208 events in 8 oocytes. Data are from oocytes injected with either AD3-B11 (*n* = 2), AD3-TEC (*n* = 3) or AD2-B11 (*n* = 3). (**B**) Results from parallel experiments injecting 10 nl of synthetic of Aβ42 oligomers (1 μg/ml). Top image is a line-scan depicting fluorescent intensity over time. Left plot is generated by superimposing traces from 10 selected events and inset is their resulting averaged profile. Center end left plots are distributions of events durations (mean duration 165.06+/−3.3 ms) and amplitudes (mean amplitude ΔF/Fo 0.47+/−0.038) measured from 297 events in 4 oocytes. (**C**) Top image depicts a line-scan measurement of fluorescent signal over time after injection of 10 nl of 30 nM solution of IP_3_. The plot below shows superimposed traces of 10 puffs and inset is the corresponding averaged profile. On the right are distributions of event durations (mean duration 142.06+/−4.9 ms) and amplitudes (mean amplitude ΔF/Fo 0.63+/−0.045) measured from 243 puffs from 4 oocytes.

**Figure 6 cells-11-03630-f006:**
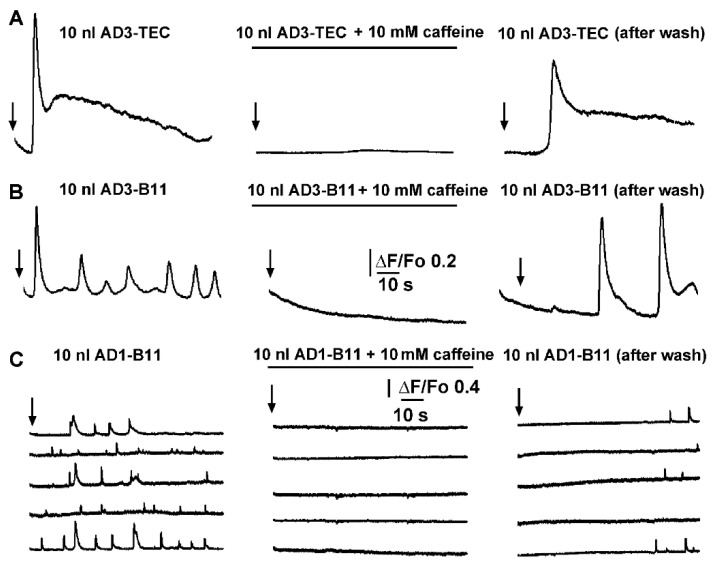
Caffeine reversibly inhibits local and global Ca^2+^ signaling triggered by intracellular injection of brain extracts from AD patients. (**A**) Left trace shows global Ca^2+^ fluorescent signal triggered by injection of 10 nl of AD3-TEC. Central panel show equivalent experiment in the presence of caffeine, and right panel after caffeine washout in the same oocyte. Black arrows indicate time of injection in each panel. (**B**,**C**) corresponding experiments in different oocytes during intracellular injection of 10 nl of AD1-B11 and AD3-B11 brain extracts, respectively. Signals in (**A**,**B**) are recorded from 40 × 40 μm regions of interest (global), whereas traces in (**C**) are from 3 × 3 pixel (1 μm^2^) regions centered on 5 different event sites within the imaged field.

**Figure 7 cells-11-03630-f007:**
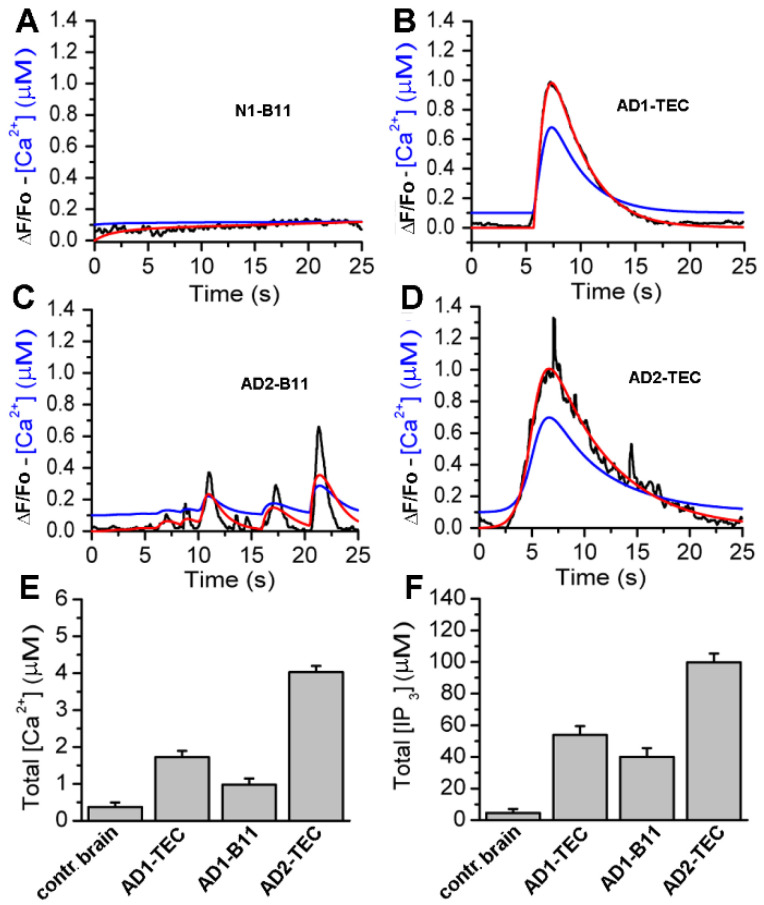
Computational analyses of the fluorescence signals allows quantification of Ca^2+^ released and the corresponding IP_3_ generated by each brain extract. In Panels (**A**,**B**), the black trace is ΔF/Fo (experiment), red is ΔF/Fo (model), and blue is the estimated Ca^2+^ from the model. (**C**,**D**), are sample traces from control brain, AD3-TEC, AD3-B11, and AD2-B11. The bar plots in (**E**,**F**) show the total concentration of Ca^2+^ released and IP_3_ generated by brain extracts in 20 s, both estimated from the model fits. The error bars represent standard deviation from three experiments.

**Figure 8 cells-11-03630-f008:**
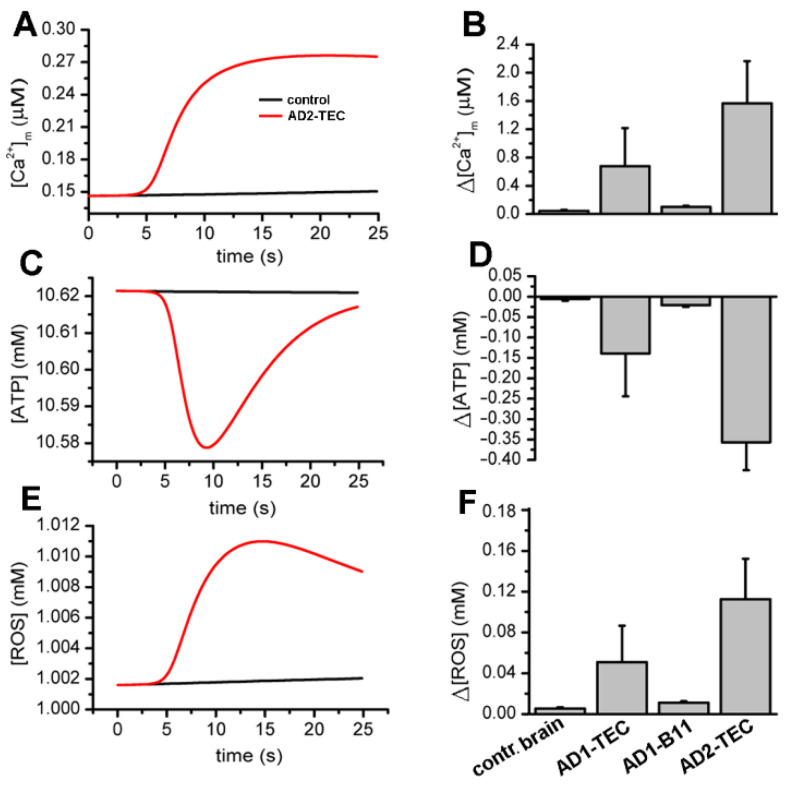
Model simulations showing rises in cytosolic Ca^2+^ due to AD brain extracts result in decreased ATP and higher ROS production as compared to an extract from control brains. Panels (**A**,**C**,**E**) display sample traces showing modeled changes in mitochondrial Ca^2+^, cellular ATP, and ROS, respectively, in oocytes injected with extract from control (black) and AD-affected brains (red). The bar plots in (**B**,**D**,**F**) show the total concentration of Ca^2+^ buffered by mitochondria, decrease in ATP, and increase in ROS in 20 sec due to brain extracts. The error bars represent standard deviation from three experiments.

**Table 1 cells-11-03630-t001:** Brain extracts tested.

	Patient ID	Age	Sex	PMI	NPDx	Tangle Stage	Plaque Stage	MMSE	Brain Area
**N1**	09-03	81	M	6.4	Normal	2	0	27	B11
**N2**	47-97	71	M	4.9	Normal	1	A	/	B11
**N3**	07-03	84	F	4.25	Normal	3	A	29	B11
**AD1**	62-98	81	M	5	AD	5	B	4	B11
**AD2**	34-99	77	F	5.4	AD	4	B	0	B11
**AD3**	04-02	83.2	M	3.5	AD	6	C	0	B11/TEC

PMI, postmortem index; NPDx, neuropathological index; MBC, mild Braak changes; AD, Alzheimer’s disease; MMSE, mini mental state examination.

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
