# Peer review of "Intracellular Injection of Brain Extracts from Alzheimer’s Disease Patients Triggers Unregulated Ca2+ Release from Intracellular Stores That Hinders Cellular Bioenergetics"

_cells, 2022, doi:10.3390/cells11223630_

Round 1
Reviewer 1 Report
This manuscript by Pensalfini et al is a follow-up study on an earlier elegant mechanistic study by the authors that analyzed the intracellular Ca2+ mobilization by synthetic Ab42 amyloid oligomers microinjected to Xenopus oocytes. Here, the authors convincingly demonstrate that the Ca2+ mobilization mechanisms and patterns identified using the synthetic amyloid oligomers can be recapitulated using AD patient brain extracts, rich in OC-immunoreactive Ab42 aggreagtes, microinjected to the oocytes. Some in silico computational attempts are given to bring the data further to a bioenergetics and ROS/redox balance/imbalance context; however, without any experimental support.
Overall, this reviewer feels that this study and the invested efforts are important contributions to the Alzheimer’s disease mechanisms field.
The study is experimentally well designed and executed except that the computational data for bioenergetics lack experimental validation. Further specific concerns are listed below in order of appearance.
1) 40 uM final cytosolic concentration for the Fluo4 dextran sounds quite high. Is it not bright enough at lower concentrations? Alternatively, was it used to isolate better the elementary release events? Please confirm the value and/or clarify the purpose.
2) What was the reason to set the extracellular pH to 7.2 instead of 7.4? Is that something frog-specific?
3) Fig.1A. besides the dot blot, it would be useful to know the size range of the oligomers that remained in the PBS soluble supernatant (e.g. run a native gel).
4) The mixing of OC antiserum with the brain extracts could be strengthened by clearing out OC-reactive proteins using a protein purification method (e.g. pulldown with immobilized OC antibodies). This might of course have feasibility issues, depending on the antibody.
5) Isn’t there a way to stimulate IP3R-mediated Ca2+ mobilization using a GPCR agonist in the Xenopus oocyte? If the suggested target of Ab42 is at the level of IP3 generation, then the demonstration that the cells still respond to caged IP3 when brain extract is injected after the OC antiserum pre-incubation will not address if IP3 generation was suppressed or not.
6) I don’t believe it is right to model bioenergetics without direct experimental support. As it is presented, the model likely gives the assumption that the transfer of cytosolic Ca2+ signals to the mitochondrial matrix is not affected by the Ad42 oligomers. Even if it was experimentally solid, the authors need to acknowledge that one of the great limitation of their model is that the cellular system is a frog egg cell, not a mammalian neuron. Metabolic circuits, bioenergetics and even Ca2+ signaling are different (e.g. many neuron types express ryanodine receptors that Xenopus oocytes apparently lack). The consequence of ectopic Ca2+ signaling activity on energetics and cell fate in one cell type can be very different from the other. This does not take away the strength of the oocyte model system in terms of the demonstration that Ab42 (synthetic or patient-derived) act on IP3 mobilization.
Minor:
- “These Ca2+ events consist of both local and global Ca2+ signals and showed strong similarity with those evoked after injection of synthetic A42 oligomers and intracellular IP3 release [15, 16].” –I believe, the authors are citing the wrong paper of their own. Instead of 15, they should use Demuro and Parker 2013 J Neurosci PMID 23447594
Reviewer 2 Report
The present work highlights the mechanism of action of endogenous Aβ oligomers altering calcium signalling in neighbouring cells. Building upon former works in which the same mechanism has been described using synthetic Aβ peptides, the major relevance of the present work derives from the use of human brain extracts derived from either normal individuals or Alzheimer disease patients.
Comments:
1) The abstract describes only the results from figures 1 to 4, while the experimental evidence of the mechanism of action of the Aβ oligomers in relation to IP3 and IP3Rs is not mentioned. Likewise, the computational modelling is not mentioned in the abstract. Therefore, the sentence “These results support the hypothesis that endogenous…” appears not enough justified in the abstract. Please revise the abstract accordingly, including all the achieved results.
2) Figures 7 and 8 are showing the results from an experimental duplicate. Is there a solid reason for not having repeated the experiments in triplicate? This would have allowed the authors to perform also the statistics which in the current state is not possible. As the data shown in fig. 7-8 are the result of computational simulations and not biological experiments, I assume they could be easily repeated.
3) A large number of typos are present throughout the text. The manuscript writing should be carefully checked before the next submission. Here below are some examples:
a. Aβ oligomers are sometimes named Abs oligomers. Sometimes only A oligomers. They all should be converted to Aβ, for consistency.
b. Comma is several times put between the subject and the verb of the sentence. All commas between subject and verb should be removed, unless they isolate an aside.
c. The reference numbers are put throughout the text in square brackets. However, in the discussion, they are present sometimes within round brackets or double brackets. Please modify them to make them consistent throughout the manuscript.
d. In figure 6A, 10 nl is reported twice as 10 ln.
e. Line 45, add “of” before Aβ42 oligomers
f. Line 65, AD brain extracts, remove “s” from brains
g. Line 144, of a specific oligomeric forms, remove “a”
h. Line 173, to consistently triggered, remove “ed”
i. Line 219, veicle, add “vehicle”
j. Line 230, are superimpose, add “d” to the verb
k. Line 231, in outlined, replace “in” with “is”
l. Line 235, originate, add “s” to the verb
m. Line 266-267, columns represents numbers, remove “s” from the verb represent
n. Line 288, localize Ca2+, add “d” to localize
o. Line 365, after to injection, remove “to”
p. Line 370, Puffs, remove “s”
q. Line 387, remove double period from the end of the sentence
r. Line 433, the word “From” is repeated three times in the same sentence and it is hard to read. Replace at least one “From” with “of” when referring to donors.
s. Line 460, Ca2+ signalling are coordinated. Replace “are” with “is”
t. Line 467, is further strength by, replace “strength” with “strengthened”
u. Line 469, the ability all, replace “all” with “of”
v. Line 472, a specific aggregated forms, remove “s” from forms.
w. Line 477, oligomers affect cells, add “ing” to “affect”
x. Line 477, overproduction IP3, add “of”
y. Lines 483-484, rephrase
z. Line 504, amount Ca2+, add “of”
Round 2
Reviewer 1 Report
The authors have adequately addressed most of the concerns raised. I have just one comment for their point#5: "We found the suggestion of using GPCR agonist as control rather than caged IP3 appropriate. However we believe that the present results provide strong evidence supporting the ability of OC-antibody to block PLC-dependent IP3 production that is otherwise consistently initiated by AD affected brain extracts when injected alone..."
Based on ref#11, there are IP3R-dependent as well as independent mechanisms for the Ca2+ mobilization by Ab42 from the IP3-sensitive ER pool. The OC antibody would likely block both pathways and lead to those large [Ca2+]c responses upon IP3 uncaging. Please, then include these coexistent pathways to the discussion/interpretation of the results.
Author Response
We thank reviewer #2 for the detailed comments and suggestions on our manuscript.
As requested, we have now added a paragraph in the discussion section ( from line 514 to line 525) about reference #11 as reported below
"In comparable conditions, e.g. pre-treatment with caffeine and intracellular delivery of Ab42 oligomers, similar results were reported using SH-SY5Y cells as a model system [11]. Furthermore, the authors observed an additional IP3-indipendent Ca2+ release from the ER, suggesting that Ab42 oligomers may trigger Ca2+ leak from the ER, which does not depend upon direct interaction with IP3Rs; a behavior that we did not observe in the time range of our experiments. However, in similar experimental conditions including the presence of zero extracellular Ca2+ and caffeine pre-treatment, we did observe cytosolic fluorescent Ca2+ transients after longer time following intracellular injection of Ab42 oligomers (unpublished data Demuro and Parker). Considering the amplitudes and temporal evolution of these Ca2+ transients we can speculate that in time, interaction of Ab42 oligomers with ER membrane may trigger formation of Ab42 pores in the ER membrane leading to Ca2+ leak from the ER described by the authors in [11]."
